# AI4HPC: Library to Train AI Models on HPC Systems using CFD Datasets

**Eray Inanc**
Forschungszentrum Jülich
e.inanc@fz-juelich.de

**Rakesh Sarma**
Forschungszentrum Jülich
r.sarma@fz-juelich.de

**Marcel Aach**
Forschungszentrum Jülich
University of Iceland
m.aach@fz-juelich.de

**Rocco Sedona**
Forschungszentrum Jülich
r.sedona@fz-juelich.de

**Andreas Lintermann**
Forschungszentrum Jülich
a.lintermann@fz-juelich.de

## Abstract

This paper introduces *AI4HPC*, an open-source library designed to integrate Artificial Intelligence (AI) models and workflows in High-Performance Computing (HPC) systems for Computational Fluid Dynamics (CFD)-based applications. Developed by CoE RAISE, *AI4HPC* addresses not only challenges in handling intricate CFD datasets, model complexity, and scalability but also includes extensive code optimizations to improve performance. Furthermore, the library encompasses data manipulation, specialized ML architectures, distributed training, hyperparameter optimization, and performance monitoring. Integrating AI and CFD in *AI4HPC* empowers efficient analysis of extensive and large-scale datasets. This paper outlines the architecture, components, and potential of *AI4HPC* to accelerate and augment data-driven fluid dynamics simulations and beyond, demonstrated by showing the scaling results of this library up to 3,664 GPUs.

## 1 Introduction

Artificial Intelligence (AI) has tremendous potential to enhance data- and compute-driven methods in domains such as Computational Fluid Dynamics (CFD). CFD simulations can be computationally intensive, thus, require High-Performance Computing (HPC) resources. They usually produce complex, large-scale datasets that need dedicated post-processing routines [1], providing many opportunities for exploiting AI-driven methodologies. In various domains, it has already become a common practice to apply AI techniques, which has shown remarkable results in the engineering realm, such as in design and optimization tasks [2]. However, AI also faces challenges, especially in demonstrating good scalability on HPC systems for compute and data-intensive tasks in processing large datasets and generating complex and high-dimensional models.

To address these challenges, CoE RAISE[1] has developed the open-source library *AI4HPC*[2] that aims to bridge the gap between CFD and AI by providing tools, methods, and code optimizations to train Machine Learning (ML) models with complex and enormous CFD datasets on large-scale HPC systems. *AI4HPC* consists of five main components:

1. Data manipulation methods (preprocessing, augmentation, normalization) for CFD datasets.

---

[1]CoE RAISE https://www.coe-raise.eu
[2]*AI4HPC* https://www.ai4hpc.com

Workshop on Advancing Neural Network Training at 37th Conference on Neural Information Processing Systems (WANT@NeurIPS 2023).

2. ML architectures (compression, generative, regression networks) for CFD use cases.
3. Interchangeable communication libraries to handle distributed training on HPC systems.
4. HyperParameter Optimizations (HPO) tool capable of scaling to large HPC systems.
5. Monitoring and performance benchmarking tool.

This paper introduces the design and implementation of *AI4HPC* and its features and functionalities. Also, the scalability of *AI4HPC* on an HPC system is presented. This paper shows that *AI4HPC* can enable efficient and effective AI training with CFD dataset on HPC platforms.

## 2 Compilation and design

*AI4HPC* is available in an open-access repository, which can be cloned with the following command.

```
git clone https://gitlab.jsc.fz-juelich.de/CoE-RAISE/FZJ/ai4hpc/ai4hpc
```

The following code snippet provides the manual to compile *AI4HPC*.

```
python setup.py --help
```

The setup script of *AI4HPC* only depends on a recent Python version (3.10 or later). At this stage, the setup script is already pre-configured for five HPC systems, i.e., for the Jülich Wizard for European Leadership Science (JUWELS) [3], Jülich Research on Exascale Cluster Architectures (JURECA) [4], the Dynamical Exascale Entry Platform – Extreme Scale Technologies (DEEP-EST) [5] systems at the Jülich Supercomuting Centre (JSC), the CTE-AMD[3] system at Barcelona Supercomputing Center (BSC), and the Large Unified Modern Infrastructure (LUMI)[4] system at IT Center for Science (CSC). Moreover, configuring *AI4HPC* for other HPC systems is trivial and straightforward.

*AI4HPC* is based on PyTorch with Python as the host language and uses the Distributed Data Parallelization (DDP) strategy to train models on large CFD datasets. DDP splits and distributes the data among various devices, e.g., Central Processing Units (CPUs), Graphics Processing Units (GPUs), or Intelligent Processing Units (IPUs). It synchronizes the model parameters using one of four open-source external libraries, i.e., the standard communication package of PyTorch (PyTorch-DDP), Horovod [6], DeepSpeed [7], or HeAT [8]. The libraries PyTorch-DDP and HeAT use the NVIDIA Collective Communications Library (NCCL) by NVIDIA or Gloo by Facebook for communication. Horovod and DeepSpeed uses Message Passing Interface (MPI) for inter-node and NCCL for intra-node communication. *AI4HPC* gives users the alternative to choose the best communication library for different systems. It also supports computation and communication overlap, meaning that gradient computation and exchange can happen simultaneously. This can speedup the training by using multiple CUDA operations. *AI4HPC* also uses an elastic launch strategy to adjust the number of devices dynamically and works with the SLURM scheduler.

*AI4HPC* uses external libraries for input, output, and post-processing. For input datasets in HDF5 format, NumPy-based methods are used and converted to PyTorch tensors. The standard PyTorch checkpointing operation saves the model state of the first device.

## 3 Components and optimizations

*AI4HPC* consists of separate modules for ML models, data loading routines, parsable variables, and post-processing routines. This way, the backbone of *AI4HPC* does not need to be modified when extending this framework for individual cases. The following two subsections describe two important modules and how they operate. This is followed by two important issues tackled by *AI4HPC*, i.e., training error definition and the large minibatch size problem. The last two subsections introduce the HPO and monitoring tools.

### 3.1 Models

*AI4HPC* includes several state-of-the-art ML architectures tailored to CFD problems. Due to the limited length of this paper, the following architectures are briefly explained: (i) Convolutional

---

[3]CTE-AMD https://www.bsc.es/innovation-and-services/technical-information-cte-amd
[4]LUMI https://www.lumi-supercomputer.eu

AutoEncoders (CAEs) are Deep Learning (DL) methods that perform dimensionality reduction using convolutional filters, which can effectively reduce the local disk space requirements of a CFD field; (ii) Convolutional Defiltering Models (CDMs) are based on Diffusion Probabilistic Models (DPMs) and are used for super-resolution tasks, generating highly-resolved CFD fields from low-resolution data; (iii) Convolutional AutoEncoder - Prediction Networks (CAE-PNs) are types of CAEs combined with a Convolutional Neural Network (CNN) to predict a quantity of interest, e.g., the total power-saving of an actuated airfoil (as in [1]) as a function of operational conditions; and (iv) Flow Transformers (FlTs) predict the next time-step of input, this way, expensive time integration methods can be replaced.

## 3.2 Data loader

*AI4HPC* consists of a specific multi-process dataloader, which can process datasets with irregular dimensions. The large input is primarily reshaped into smaller patches, where each patch is treated as a batch. This way, irregular CFD grids (e.g., non-equidistant meshes) can be handled effortlessly. For faster input, the data transfer between the CPU (as host) and the GPU (as device) uses page-locked (or pinned) memory; thus, `cudaMemcpy` operations can be skipped.

## 3.3 Training error

The training error in CFD problems is usually based on predicting a tensor field instead of a classifiable target. To account for this, *AI4HPC* uses a training error computed using the Mean-Squared Error (MSE) between the reference and the reconstructed fields [9]. As each device in the DDP strategy has its own share of training data resulting in corresponding training errors, *AI4HPC* averages the error across all devices using an `allreduce` operation.

## 3.4 Large minibatch size problem

Two approaches to circumvent the issue of large minibatch sizes are implemented in *AI4HPC*. Briefly, this problem occurs as each device has its own batch, named microbatch. The total minibatch size of the training becomes the sum of the microbatch sizes over the total number of devices. Therefore, the minibatch size increases since training data is distributed across multiple resources in data parallelism, impacting the model's accuracy [10]. *AI4HPC* provides solutions to this problem by using either (i) the Adaptive Summation (AdaSum) algorithm [11] or (ii) scaling the learning rate with the number of devices.

## 3.5 Hyperparameter Optimization tool

Optimizing the hyperparameters is an important aspect of an ML training. *AI4HPC* includes a scalable HPO tool that optimally performs such tasks using HPC resources. This is done by the Ray Tune library[5] that features a smooth integration of PyTorch-based training scripts and enables two stages of parallelism: (i) run each trial (model with different hyperparameters) in parallel on multiple GPUs using the DDP strategy, and/or (ii) run several trials in parallel on an HPC system (via Ray Tune itself).

## 3.6 Monitoring tools

The performance of *AI4HPC* can be monitored using NVIDIA's system-wide performance analysis tool Nsight API[6] and PyTorch's standard profiler. *AI4HPC* also prints important performance metrics to the standard output file, such as the epoch runtimes, loss, or memory print. GitLab's CI routines are also implemented that continuously compile and benchmark *AI4HPC*.

# 4 Benchmarks

The benchmarking tool of *AI4HPC* is run on JUWELS up to 916 compute nodes, which is equivalent to parallel usage of 3,664 NVIDIA A100s[7]. The scaling behavior results are shown in Fig. 1. Here,

---

[5]Ray Tune `https://docs.ray.io/en/latest/tune/index.html`
[6]Nsight API `https://developer.nvidia.com/nsight-perf-sdk`
[7]NVIDIA A100 `https://www.nvidia.com/en-us/data-center/a100`

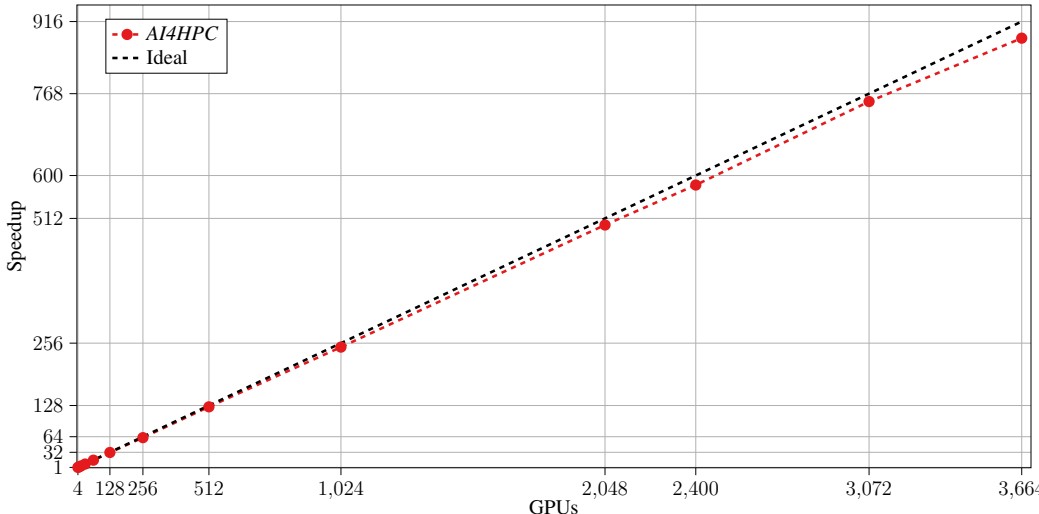

Figure 1: Results of the benchmarking tool of *AI4HPC* run on JUWELS up to 916 compute nodes (3,664 NVIDIA A100s). The U-Net architecture with over 52 million trainable parameters that occupies 36 GB memory is trained with synthetically created CFD dataset. The black-dashed line depicts the ideal speedup.

U-Net architecture [12] for super-resolution purposes with over 52 million trainable parameters that occupy 90% of available GPU memory per device (36 GB / 40 GB) is trained with synthetically created CFD dataset (i.e., input/output routines are disabled) using AI4HPC with Horovod's communication backend. The benchmark shows that a speedup of 881.92 (maximum theoretical speedup is 916) is achieved with 3,664 GPUs in parallel, where the training with 4 GPUs is used as the baseline. This leads to a scaling efficiency higher than 0.96. Note that this scaling performance is only possible with a significantly large dataset. Using small datasets would suffer from a communication bottleneck due to insufficient data per device.

Moreover, the code optimizations implemented in *AI4HPC* to leverage the HPC system's performance drastically reduce the training runtimes. Current tests show that a factor of 10 speedup is achieved compared to the non-optimized frameworks (not shown for brevity).

## 5 Conclusion

This paper presented *AI4HPC*, which empowers researchers and engineers to effectively harness AI techniques for complex CFD datasets on HPC systems. The components of the library include data manipulation methods, ML architectures, communication libraries, HPO techniques, and performance monitoring tools, which collectively contribute to a holistic solution.

The successful implementation and deployment of *AI4HPC* are demonstrated through its integration with leading HPC systems and its compatibility with diverse communication libraries. This interoperability optimally utilizes the capabilities of various hardware devices, enabling efficient device management. Furthermore, *AI4HPC*'s ability to handle non-equidistant CFD grids and training errors unique to CFD problems showcases its adaptability and sophistication.

The presented benchmarks on the JUWELS system highlight *AI4HPC*'s scalability and performance enhancements. The remarkable speedup achieved on up to 3,664 GPUs with a scaling efficiency of 0.96 emphasizes the ability of the library to leverage the computational resources efficiently, ultimately leading to a substantial reduction in training runtimes.

In essence, *AI4HPC* emerges as a pivotal contribution to the fields of AI, CFD, and HPC. As these fields continue to evolve, *AI4HPC* holds the potential to play a transformative role in driving cutting-edge research and applications in compute- and data-driven fluid dynamics and beyond.

## Acknowledgements

The research leading to these results has been conducted in the CoE RAISE project, which receives funding from the European Union's Horizon 2020 – Research and Innovation Framework Programme H2020-INFRAEDI-2019-1 under grant agreement no. 951733. We acknowledge the EuroHPC Joint Undertaking for awarding this project access to the EuroHPC supercomputer LUMI, hosted by CSC (Finland) and the LUMI consortium.

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
