# OpenReview forum: "AI4HPC: Library to Train AI Models on HPC Systems using CFD Datasets"
_NeurIPS.cc/2023/Workshop/WANT — WANT@NeurIPS 2023 Poster_

### Official Review · Reviewer_5Lh7 · 2023-10-20
**No or very little contribution with questionable presentation of the library**

**Rating:** 2
**Confidence:** 4

**Review:**

Overview

The paper introduces an open-source library that aims to connect artificial intelligence systems and high-performance computing for exascale applications. The authors describe the components of the library, all being external libraries, which enable to manipulate data, train machine learning algorithms, handle distributed training, optimise training hyperparameters, and, lastly, monitor and benchmark the progress. The paper states the scaling results up to 3664 GPUs.

Strengths:
- Good intentions and starting point, more like a proof of concept

Weaknesses:
- No explanation of how the library helps a user, who for example can come from a physics field, to apply AI to their problem in HPC context. How flexible is the library to fulfil the needs of a user? How much work would one perform to use the library for their specific dataset and deep learning model, for example, lately popular for CFD-based problems PINNs models?
- No mention of any other existing library which performs a similar function that would explain the need for the presented library.
- The plot presented (figure 1) is just for one of the three supercomputers, why others are not included?
- No description of mentioned implemented optimizations or the experiment details where the speedup is achieved, motivated by "not shown for brevity". There was a possibility to submit a paper of up to 9 pages.
- Paper style: (1) "citing" of software should be done with usual citations (usually given at software homepages), not footnote links, (2) commands to clone and compile a library usually are not included in a paper, those are left in the documentation page that can be mentioned in a paper (3) lack of citations here and there.
- Since the paper presents a library, the code could have been provided for a review, e.g., with the use of Zenodo anonymously.

Conclusion: The paper lacks contributions to *advancing* neural network training or related topics, which are included in the workshop. The submission with improvements suits more the Journal of Open Source Software (JOSS).

---

### Official Review · Reviewer_H7xq · 2023-10-23
**More details are required to improve this paper**

**Confidence:** 3

**Review:**

This paper presents a library using AI to accelerate Computational Fluid Dynamics (CFD) simulations. The library is based on PyTorch and exhibits an impressive scalability up to 916 compute nodes (3664 GPUs). Authors identified several challenges when combining AI and CFD and propose solutions for each of them.


## Strengths

- description of challenges with several solutions for each of them
- performance results seem good

## Weaknesses

- it is not clear what AI is actually optimizing in CFD simulations
- for each challenging point, different solutions are proposed, but the pros and cons or the impact of each solution on performance are not discussed
- there is no performance comparison with state-of-the-art competitors, making it difficult to assess whether the library actually provides a performance improvement
- there is no performance analysis: what makes the library so efficient?


## Recommendations

- section 2: lines 36 to 41 are useless, mentioning the website of the library is enough, instructions to install the software should be in the documentation (or in an appendix)
- section 2, lines 52-53: "This can speedup the training by using multiple CUDA operations." what does it mean? using *simultaneously* CUDA operations?
- section 3, lines 62-63: "The following two subsections describe two important modules and how they operate." which modules?
- section 3, line 67: "LIBRARY includes several state-of-the-art ML architectures tailored to CFD problems." some references to the state-of-the-art would be welcome
- section 3, lines 107-108: mentioning the library uses GitLab's CI is useless
- section 4: which optimizations were used for the execution on figure 1? (for instance: which communication library was eventually used to evaluate the performance?)

The paper would probably benefit from a longer version. This short version raises many problems, explain (too) briefly how to solve them, but it lacks of details. Describing the problems and solutions more in details, evaluating the impact on performance would bring more value to the paper.

---

### Official Review · Reviewer_gjx4 · 2023-10-24
**This paper presents an open source library for running AI models using CFD datasets on high performance systems. The main contributions are:  -Integration of AI and CFD. - IA Code Optimizations. - Encompasses various features such as data manipulation, specialized ML architectures, distributed training, hyperparameter optimization, and performance monitoring. -  Resource scability.**

**Confidence:** 3

**Review:**

This paper introduces the development of a library for AI models, taking into account all the aspects involved in the preprocessing and execution of CFD application-oriented models. The contribution of each aspect is explained as follows:
- Models: includes different architecture models such as convolutional networks, probabilistic diffusion, prediction and transformers.
- Data loader: features a multi-process loader that allows processing of irregular dimensions. The main contribution is that it allows to reduce the number of cudaMemcpy.
- Training error: is calculated using mean-squared error (MSE). Since the training data are distributed over all devices, the allreduce operation is used to obtain the final average error.
- Large minibatch size problem: to solve this problem uses the adaptive summation algorithm (AdaSum) and learning rate scaling with the number of devices.
- Hyperparameter optimization tool: use the RayTune library in Pytorch which provides two functionalities:
              * Run each model with different hyperparameters in multiple GPUs using DPP strategy.
              * Run multiple models in parallel on an HPC system.
- Monitoring tools: use NVIDIA system-wide performance analysis tool Nsight API and Pythorch profiler.
They benchmarked the library using up to 916 compute nodes where each node has 4 GPUs. They took 1 node as baseline and showed that the speed up remains almost constant as we increase the number of nodes. They clarify that it is necessary to use a very large dataset so that the communication/synchronization of the nodes is not predominant. The result is an efficiency of 0.96 using 916, which is very good.

Strenghts:
- Is a work of great importance for AI model developers.
- Theoretically, it will speed up the CFD model training process.
- Is applicable to a wide range of ML architectures.
- The compilation of the library is easy and has a help menu to know all the compilation options.

Weaknesses:
- The article is very limited.
- The introduction does not mention whether there are already other libraries that perform the same or similar tasks.
- It does not present a clear explanation of which are the improvements applied in each module of the library.
- The library is limited to CFD dataset algorithms only. How does it behave with other datasets?
- The design section does not present different subsections for each library module. It mentions everything together, which is not very clear.
- The results do not mention which ML architecture was used for the tests.
- It does not explain how the test is launched to obtain results.
- It presents incomplete results, only a graph with the speedup.
- It does not mention what the future work is. Test with other models? with other datasets? add optimizations?

Clarity:
 - The article is well written. The "benchmarks" section I would call "results".
 - In section 4, I would mention at the beginning that each node has 4 gpus.

Relation to prior works:
- There appears to be no work history. This work presents the library for the first time.
- The library can contribute to the training of models with large datasets such as CFD and there is currently no similar.

---

### Official Review · Reviewer_598f · 2023-10-25
**Library integrating multiple tools to leverage HPC resources to train DL models on CFD**

**Confidence:** 5

**Review:**

The paper presents a library to train deep learning architectures on CFD datasets by leveraging HPC resources.

Such a library is much welcomed as there is a need to automate the training of Artificial Intelligence for Science, especially in the field of computational fluid dynamics. The paper is easy to read. However, it presents the following flaws:

- The contribution is unclear. Most of the library features are already provided by existing tools, with which deep learning practitioner are familiar. Is the library merely a patchwork of these tools, whose coordination represents nonetheless substantial engineering work, or does implement new techniques to tackle the problems arising with the training of models for CFD on HPC? The contribution should be made clearer. Perhaps, the engineering work would be better suited for publications in the [Journal of Open Source Software](https://joss.theoj.org/).
- The paper lacks of comparison with existing libraries and frameworks. For instance, [PhiFlow](https://github.com/tum-pbs/PhiFlow), [Modulus](https://developer.nvidia.com/modulus), and [PDEBench](https://arxiv.org/pdf/2210.07182.pdf) already provide collections of datasets and models dedicated to CFD. How can these tool integrate with the presented library to generate data or train a model? Moreover, there already exist frameworks that leverage HPC resources to train models for science or CFD (e.g. [Scalable HPC & AI Infrastructure for COVID-19 Therapeutics](https://dl.acm.org/doi/abs/10.1145/3468267.3470573), [Training Deep Surrogate Models with Large Scale Online Learning](https://proceedings.mlr.press/v202/meyer23b/meyer23b.pdf), [Enabling Machine Learning-Ready HPC Ensembles
with Merlin](https://arxiv.org/pdf/1912.02892.pdf)). How does the library differ from these frameworks?
- L.79 reads "large input is primarily reshaped into smaller patches, where each patch is treated as a batch". Does it mean the library can only serve data to models that ingest as inputs regular patches (e.g. CNN or transformers)? What about irregular meshes and graph neural networks?
- In l.112 it is not clear if all the dataset fits in the GPU's memory along with the model ("input/output routines are disabled"). It would be appreciated to test the library in different settings, especially when I/O becomes a problem.
- Still in l.112, the ML architecture is not specified. It would be relevant to test the library on existing models and datasets to evaluate its performances.
- The source of the dataset is not specified. It is thus unclear if the library can provide dataset or create new ones by running CFD simulations. If the library is agnostic of the dataset, it seems it could be used beyond CFD. In such a case, the versatility of the library would be appreciated by a large community.
- In general, the paper rushes the description of implementation. How the different tools are coordinated together? Conversely some details could be shorten like common installation procedures (l.36) and listing in lengthy footnotes the existing blocks it relies on.

The idea and the outcome of the library is necessary for the field to progress in training better models for CFD. However, the paper lacks comparison with references and clear description of the contribution to meet the quality standard for acceptance. I am confident that a rewriting focusing on the novelty while presenting the result of training existing models with the library would greatly benefit the paper.

---

### Meta-Review · Area_Chair_eVJU · 2023-10-25

**Recommendation:** Reject
**Confidence:** 3

**Metareview:**

While the intentions of the paper are very welcome and results seem very strong, the article lacks the necessary depth. I would encourage the authors to revise the paper adding more details and resubmitting the paper.

---

### Decision · Program_Chairs · 2023-10-28

**Decision:**

Accept (Poster)

**Comment:**

After a thorough evaluation of the paper and the feedback provided by reviewers and a meta-reviewer, we have made the decision to accept the paper. Our decision is motivated by the paper's significant relevance to one of the core topics of the workshop, which revolves around the scaling up of AI model training using multiple GPUs for scientific applications. The program chairs believe that the paper presents valuable results that are pertinent to the workshop's objectives, and we are eager to foster discussions and research advancements in this area. However, we must stress the imperative need for substantial improvements in the paper's presentation to fully realize its potential impact. Please take into account all reviewers' suggested improvements. Congratulations and hope to see you in person at the workshop and brainstorm on efficient training research together!